

# Genetic loci determining potato starch yield and granule morphology revealed by genome-wide association study (GWAS)

Vadim K. Khlestkin[1,2], Tatyana V. Erst[1], Irina V. Rozanova[1,3], Vadim M. Efimov[1,4] and Elena K. Khlestkina[1,3]

[1] The Federal Research Center Institute of Cytology and Genetics SB RAS, Novosibirsk, Russia
[2] Russian Research Institute of Farm Animal Genetics and Breeding—Branch of the L.K. Ernst Federal Science Center of Animal Husbandry, Saint-Petersburg, Russia
[3] The N.I. Vavilov Federal Research Center All-Russian Institute of Plant Genetic Resources (VIR), Saint-Petersburg, Russia
[4] Novosibirsk State University, Novosibirsk, Russia

Corresponding author
Vadim K. Khlestkin,
dir2645@yandex.ru

## ABSTRACT

**Background:** It is well-documented that (bio)chemical reaction capacity of raw potato starch depends on crystallinity, morphology and other chemical and physical properties of starch granules, and these properties are closely related to gene functions. Preparative yield, amylose/amylopectin content, and phosphorylation of potato tuber starch are starch-related traits studied at the genetic level. In this paper, we perform a genome-wide association study using a 22K SNP potato array to identify for the first time genomic regions associated with starch granule morphology and to increase number of known genome loci associated with potato starch yield.

**Methods:** A set of 90 potato (*Solanum tuberosum* L.) varieties from the ICG "GenAgro" collection (Novosibirsk, Russia) was harvested, 90 samples of raw tuber starch were obtained, and DNA samples were isolated from the skin of the tubers. Morphology of potato tuber starch granules was evaluated by optical microscopy and subsequent computer image analysis. A set of 15,214 scorable SNPs was used for the genome-wide analysis. In total, 53 SNPs were found to be significantly associated with potato starch morphology traits (aspect ratio, roundness, circularity, and the first bicomponent) and starch yield-related traits.

**Results:** A total of 53 novel SNPs was identified on potato chromosomes 1, 2, 4, 5, 6, 7, 9, 11 and 12; these SNPs are associated with tuber starch preparative yield and granule morphology. Eight SNPs are situated close to each other on the chromosome 1 and 19 SNPs—on the chromosome 2, forming two DNA regions—potential QTLs, regulating aspect ratio and roundness of the starch granules. Thirty-seven of 53 SNPs are located in protein-coding regions. There are indications that granule shape may depend on starch phosphorylation processes. The *GWD* gene, which is known to regulate starch phosphorylation—dephosphorylation, participates in the regulation of a number of morphological traits, rather than one specific trait. Some significant SNPs are associated with membrane and plastid proteins, as well as DNA transcription and binding regulators. Other SNPs are related to low-molecular-weight metabolite synthesis, and may be associated with flavonoid biosynthesis and

circadian rhythm-related metabolic processes. The preparative yield of tuber starch is a polygenic trait that is associated with a number of SNPs from various regions and chromosomes in the potato genome.

# INTRODUCTION

Starch is one of the most important renewable and economically notable organic resources of humankind (*Khlestkin, Peltek & Kolchanov, 2018*). Simplicity of potato growing and ease of industrial production of pure potato starch makes it a necessary component and starting compound in food and chemical industries and determines valuability of an exhaustive study of all aspects of manufacturing and commercial application of potato starch is necessary. There are many publications related to chemical and biochemical transformations of starch of diverse botanical origin and differences properties of various starches. Surprisingly, little attention is paid to difference in starch properties at the level of different cultivars, such as potato cultivars. Commercial starch is still subdivided as "potato starch", "rice starch", "corn starch" and so on, despite it is well-known that there are significant differences in the properties of starch of the same botanical origin. From general considerations, it is evident that chemical and biochemical reaction ability of the raw potato starch must depend on its granules crystallinity and morphology. If so, starches from different potato varieties with different granules' properties should fit different applications better.

(Bio)chemical reaction capacity of raw potato starch depends on the crystallinity, morphology and other chemical and physical properties of starch granules. For example, after chemical acylation of the raw potato starch and fractionation of resulting acetylated starch according to granule size, amylose and amylopectin may be isolated and characterized by degree of substitution (DS) and degradability with α-amylase, β-amylase and amyloglucosidase. In contrast to amylose, the DS of the amylopectin from the differently sized granules increased with decreasing granule size. The acetyl groups of the amylose molecules from small granules are more heterogeneously distributed and located more closely to the non-reducing ends compared to amylose from larger granules. The amylose populations from small granules of the acetylated starches were less susceptible to all the enzyme degradation reactions than the amylose from the large granules, even though the DS was similar. Additionally, the acetyl group distributions were different for amylopectin from different granule size fractions (*Chen, Schols & Voragen, 2004*; *Chen et al., 2005*). Some benefits of small potato starch granules are summarized in European patent (*De Vetten & Heeres, 2004*). The large size of the potato granules often hampers the utilization of potato starches, for example, in printing ink formulations, wherein potato starch granules may obstruct the printing apertures. Smaller granules facilitate chemical modification of the starch, since the surface-volume ratio is important

for the accessibility of the modifying compound to the starch chains. In textile printing, for instance, carboxymethylated and roll dried potato starch of small granule size enables the production of finer prints. Furthermore, in adhesives, particularly in highly concentrated bag adhesives, crosslinked fine potato starches may be used as fillers and in drilling fluids, crosslinked and modified fine starches are expected to reduce fluid loss. In addition, smaller granule potato starches may be used as filler/structurant in soap, since this starch provides a pleasant sensation to the skin. Small granular potato starches are particularly suitable for applications in the food industry. Small granular potato starch provides such advantages as lower starch dosage level, better taste profile, and smooth but not excessively swollen granules (*Singh & Kaur, 2004*). Other practical and important properties that are sensitive to starch granule size are liquid composite viscosity (*Zhou et al., 2002*), chemical modification and flowability (*Wang et al., 2016*).

It is evident that the shape of potato starch granules is closely related to genes function. Amylopectin polysaccharide is predominantly responsible for the smooth granule morphology. Even low-amylose starch granules still maintain a smooth shape, whereas starch with reduced amylopectin content (by antisense knockout of *SBE* and *GWD* genes) is associated with more oddly fissured granules (*Blennow et al., 2003*). The role of starch synthases and related genes (*GBSS, SSI–SSIII*) in potato starch granule morphology has been discussed elsewhere (*Ball & Morell, 2003*). In general, the role of certain genes in starch biosynthesis and starch granule morphology in particular is summarized in the review (*Khlestkin, Peltek & Kolchanov, 2017*).

Traditionally, the preparative yield of potato tuber starch is the only starch-related trait generally accepted as important for potato breeding. In the past decade, starch phosphorylation and amylose/amylopectin content were also added to the short list of the traits studied at the genetic level by modern molecular biology methods. Thus, forward genetic tools, such as QTL analysis (*Carreno-Quintero et al., 2012*; *Werij et al., 2012*) and association mapping (*Carpenter et al., 2015*), were used to reveal loci associated with potato traits. The tuber starch characteristics, such as starch granular size and shape, and chemical-thermal properties of 21 potato varieties were determined and associated with genetic diversity through SSR markers. SSR-based cluster analysis revealed that varieties with interesting quality attributes were distributed among all clusters and subclusters, suggesting that the genetic basis of analyzed traits might differ among the varieties (*Werij et al., 2012*).

Recently developed 22K SNP potato array is characterized by a high average density of markers, one locus per 40 kb (in the abovementioned studies (*Khlestkin, Peltek & Kolchanov, 2017*; *Carreno-Quintero et al., 2012*; *Werij et al., 2012*; *Carpenter et al., 2015*), this value does not exceed one marker per 4 Mbp). We used the chip to identify eight novel genomic regions on the chromosomes 1, 4, 5, 7, 8, 10 and 11 associated with starch phosphorylation. Some of the identified SNPs were located in noncoding genomic regions (*Khlestkin et al., 2019*).

In this article, for the first time we perform a genome-wide association study using a 22K SNP potato array to locate genomic regions associated with starch granule

morphology and to increase the number of known genomic loci associated with potato starch yield.

## MATERIALS AND METHODS

### Plant material

The same set of 90 potato (*Solanum tuberosum* L.) varieties from the ICG "GenAgro" collection (Novosibirsk, Russia) described in *Khlestkin et al. (2019)* was used in this paper as well. Complete list of the accessions used in the study and their origin is presented in Supplemental Data S1 (Excel file). All plants are grown during the period May to October 2017 in the same field in Novosibirsk (54°52′ N and 83°00′ E). Seed tubers of all cultivars were planted in two rows with 0.75 m spacing and 0.3 m distance between the plants on the rows. In total, 10 plants were planted in the row; the length of each row was 10 m. Each cultivar was planted in three replicates; distances between the replicates' plots were 2 m. Sowing: the first decade of May. Harvesting: the third decade of September. After harvesting tubers were stored for three weeks at +4 °C before starch isolation.

## STARCH ISOLATION

Potato starch was isolated from tubers according to the typical procedure described elsewhere (for example, see *Khlestkin & Erst (2017)*).

## DNA ISOLATION AND GENOTYPING

DNA was isolated from tuber skin using DNeasy Plant Mini Kit (Qiagen, Hilden, Germany) according to the standard procedure. A set of 15, 214 (71.7%) scorable SNPs (*Khlestkin et al., 2019*) was used for GWAS analysis. Genotyping information is available in Supplemental Data S2 (Excel file).

## MICROSCOPY AND IMAGE PROCESSINGS

Sample preparation, microscopic image acquisition and processing were performed according to a previously developed procedure (*Khlestkin & Erst, 2017*). Five milligrams of raw starch were suspended in one mL of distilled water and dyed while shaking with 50 μL of iodine solution. A total of 20 mL of suspended dyed raw starch granules were placed on a microplate and covered with glass.

At least four pictures of every sample were acquired (250–300 granules in every image) in transmitted light mode with bright-field technique. Micro images of starch granules obtained with the research optical microscope Axio Scope A1 (Carl ZEISS), objective—A-Plan 10x/0.25, CCD camera—AxioCam ICc 3, adaptor—TV 2/3″C 0.63x, software ZEN, total magnification 10 (objective) × 10 (ocular) × 0.63 (adaptor).

Automatic image processing and analysis were performed in the freely distributed ImageJ program. Seven morphological parameters were analyzed (Table S1; Fig. S1).

### Principal components and 2B PLS analysis

Two sets of principal components were calculated for both phenotypic morphological traits of starch granules, as well as for the genotyping data for potato varieties, through the

distance matrix using the JACOBI 4 software package (*Polunin, Shtayger & Efimov, 2014*). To calculate the distance matrix between the potato varieties, we recoded the tetraploid potato genome from four-letter codes to numerical codes, taking into account the dose of a certain allele. After the recoding, 0 was assigned to the effector allele, and 1 was assigned as the non-effector allele, and their intermediate forms were coded as 0.75, 0.5 and 0.25. For example, the AAAA allele is reflected as 1, AAAG—as 0.75, AAGG—as 0.5, AGGG—as 0.25 and GGGG—as 0.

Both sets of principal components were applied as blocks for two-block partial least squares (2B-PLS) analysis, where the first block related to phenotypic traits and the second block related to genotypic.

Population structure matrix (Q-matrix) and the genotyping data were analyzed by Bayesian cluster analysis in STRUCTURE v.2.3.4 (*Falush, Stephens & Pritchard, 2007*). Cluster analysis of the same population has been previously discussed in the authors' previous article (*Khlestkin et al., 2019*).

### Association analysis

Association analysis was performed with the TASSEL 5 package (*Bradbury et al., 2007*). Four different statistical models were tested to identify significant marker associations with potato starch yield and granule morphology: (1) general linear model (GLM) without taking into account population structure, (2) GLM using a Q-matrix of population membership (GLM+Q) taking into account the population structure, (3) GLM taking into account population membership estimates derived from principal components analysis (GLM+PCA) and (4) a composite approach that combines both Q-matrix and the average relationship between individuals or lines (null matrix), represented in TASSEL as mixed linear model (MLM). Adaptation of MLM for GWAS has been discussed in *Zhang et al. (2010)*. But MLM approach did not result into valuable results and no significant SNPs were identified with this tool.

To identify significant SNPs, two corrections were used: (i) the Bonferroni correction, where the significant threshold (0.05) is divided by the total number of tests; in this work, the total number of markers (15,214) yields a threshold of $3.29 \times 10^{-6}$ and (ii) the false discovered rate (FDR) (*Benjamini & Hochberg, 1995*), which was calculated for each isolate in each model. Some markers did not exceed the threshold but still possessed low $p$-value ($>10^{-4}$). We referred to the markers as "suggestive". The percentage of random was <10%. Belonging of identified SNPs to genes and their association with certain proteins were confirmed on the site https://plants.ensembl.org.

## RESULTS AND DISCUSSION

### Phenotyping

Our initial study (*Khlestkin & Erst, 2017*) on potato tuber starch granule morphology provided a reliable method for granule shape evaluation. This work also indicated that granule shape is specific for different potato varieties and represents a set of quantitative traits that may be applied in modern potato breeding for production starch best suitable

for industrial processing. Thus, 90 varieties of potato were harvested in 2017, and 90 samples of raw tuber starch and related DNA were isolated.

Seven morphological parameters were automatically captured and evaluated for the 90 varieties: area (area of granule projection in microscope visible bright field), Feret's diameter, minimal Feret's diameter, aspect ratio (AR), roundness, circularity and solidity. Exact explanations and formulas for the parameters are provided in Table S1 and Fig. S1.

The preparative yield of starch was also captured. It varied significantly from 7.4% (variety Agata) to 18.8% (variety Tango) (Supplemental Data S1, Excel file).

The process of capturing starch micro images, treatment and evaluation were discussed in our previous study (*Khlestkin & Erst, 2017*). For GWAS, we used average values of all the morphological parameters for every cultivar.

## Genome-wide association study

### Principal components analysis and 2B PLS analysis

"Phenotype–genotype" covariation was calculated as a set of linear bicomponents (Table S2). It was shown that the first three linear bicomponents capture 92.7% of the total covariation. All the phenotypic traits studied, as well as preparative yield, correlate with the first bicomponent. The second component showed correlations with morphology traits and no correlations with the "preparative yield of starch". "Preparative yield of starch" is the only trait that correlates significantly with the third bicomponent (Table S3). Plotting of the first and the third pairs of bicomponents showed good positive correlation between genotypes and phenotypes in both planes (Fig. S2). The haplotypes most interesting for further selection are situated at the opposite corners of the plots and are highlighted by the circles (Fig. S3). Despite the close values of the "Preparative yield of starch" in varieties Udacha and Svitanok Kievsky, they are genetically highly different.

Analysis of genotypic data gives three clusters (Fig. S4). The first and the second clusters comprise single trait—preparative yield of starch and aspect ratio, respectively. All other morphological traits studied formed the third cluster. Preparative yield of starch and aspect ratio are opposite traits on the plate. Therefore, the breeding process for optimizing the traits should be performed in opposite directions in genetic coordinates.

### Quantile-quantile plots

For all the traits studied, we evaluated whether the analysis of the population structure possesses additional accuracy in finding significant SNPs. First, the GLM analysis data were compared in the QQ-plots (quantile–quantile plots). They show that only for traits "circularity" and "Feret's diameter" one may expect correct evaluation of noticeable SNPs with the GLM method. For the other traits, GLM demonstrates inflation (overestimated $p$-values) of noticeable SNP evaluation (Fig. S5).

The GLM+PCA model takes into account the population structure and returns results of significantly higher quality (Fig. S6): calculated p-values are closer to the expected ones even in the region of high values. Thus, calculated data for "Circularity" and "Area" traits are close to theoretical ones, for "Solidity", "Feret's diameter" and "Minimal Feret's diameter", ($-\lg p$)-values are somewhat lower. For other traits, obtained ($-\lg p$)-values are
slightly higher than expected. A similar QQ correlation for the first PLS bicomponent, which includes all the genotypic traits, shows significant deviations from theoretical data and thus a lower quality of $p$-value evaluation (Fig. S6).

To account for the population structure, the GLM+Q statistical model was applied. The model worked well for "Feret's diameter", "Minimal Feret's diameter", "Circularity" and "Area" traits giving a good conformity, but did not give $(-\lg p)$-value > 4, which indicates the absence of significant markers. The "Preparative tuber starch" and "Roundness" traits showed a number of false positive SNPs (Fig. S7).

### Manhattan plots

In total, 53 significant SNPs were associated with morphological and starch yield traits (Table 1).

Recently, several genome association studies with starch granules size distribution were published for maize (*Liu et al., 2018*), wheat (*Li et al., 2017*; *Chia et al., 2020*) and Japonica rice (*Biselli et al., 2019*). In the present study, no noticeable SNPs with $p$-values higher than Bonferroni or FDR levels were found for average values for the "Area", "Feret's diameter", "Minimal Feret's diameter" and "Solidity" traits. This result is surprising, since three of the traits are related to the starch granule size, which is expected to be genotype-dependent. Indeed, if we apply ANOVA to the granules' sizes of 90 varieties involved in the study, we see that factor "variety" clearly influences the size-related traits ("Area", "Feret's diameter" or "Minimal Feret's diameter"). Building a "tree" of varieties' starch granule size identifies five clusters of varieties (genotypes) with relatively close granule size, and ANOVA within the clusters enables us to conclude that the size of the granules in the clusters is dependent on the genotype (Fig. S8).

It appears that granule size is not single- or oligogenic but is a polygenic trait. To determine the SNP pattern associated with this trait, analysis of p-values and their comparison with Bonferroni or FDR levels is not sufficient, and more advanced and complicated analysis is warranted.

### Preparative tuber starch yield

Association of SNPs with phenotypic data for the "preparative tuber starch yield" trait revealed 10 significant SNPs when the GLM model was applied. The $p$-value of one SNP exceeded the Bonferroni level, and the p-values of the other nine exceeded the FDR (Fig. 1). The SNPs were assigned to chromosomes 4, 5, 6, 7, 9 and 11.

For the "preparative tuber starch yield" trait, GLM+PCA appeared to be the best model according to the QQ plot (Fig. S6), but for the model $(-\lg p)$-values are lower than for GLM (Fig. 1). No significant SNPs were revealed, but two SNPs on chromosomes 4 and 5 may be referred to as "suggestive". Nevertheless, the GLM+Q model confirmed a significant SNP on chromosome 4. Some significant regions on chromosomes 4, 5, 9 and 11 were revealed with GLM and confirmed by GLM+Q (Table 1). Taking into account possible false-positive SNPs predicted by the QQ-plot, we lowered the FDR level following the suggestion of *Chan, Rowe & Kliebenstein (2010)*. Setting the FDR level at the 10.05 percentile of $p$-values gave a reasonable number of detectable SNPs (Table 1).
**Table 1 Significant SNPs associated with preparative yield of starch and morphological traits.**

| N | SNP | Statistical model | Significance levels | Chr | Position | *p*-Value* | Polymorphism | QTL effect | Heritability value | Trait |
|---|---|---|---|---|---|---|---|---|---|---|
| 1 | PotVar0026637 | GLM | Bonferroni | 4 | 282,828 | 2.5E−06 | T/G | 25.34766 | 0.22363 | Preparative yield |
|  |  | GLM+Q | Bonferroni | 4 | 282,828 | 2.3E−06 | T/G | 25.73074 | 0.23036 | Preparative yield |
|  |  | GML+PCA | Suggestive | 4 | 282,828 | 4.8E−05 | T/G | 18.41815 | 0.14736 | Preparative yield |
| 2 | PotVar0098904 | GLM+Q | FDR (0.05) | 4 | 61,699,989 | 2.2E−05 | T/C | 20.20908 | 0.19196 | Preparative yield |
| 3 | PotVar0098903 | GLM+Q | FDR (0.05) | 4 | 61,700,041 | 1.8E−05 | A/G | 20.66912 | 0.19391 | Preparative yield |
| 4 | solcap_snp_c2_32042 | GLM | FDR (0.05) | 4 | 67,130,719 | 3.9E−06 | A/G | 24.29934 | 0.21832 | Preparative yield |
|  |  | GLM+Q | FDR (0.05) | 4 | 67,130,719 | 4.74E−06 | A/G | 23.93846 | 0.21998 | Preparative yield |
| 5 | solcap_snp_c2_52081 | GLM+Q | Suggestive | 5 | 1,883,165 | 3.8E−05 | A/G | 18.94988 | 0.17446 | 1st bi-component |
|  |  | GLM+Q | FDR (0.05) | 5 | 1,883,165 | 1.2E−05 | A/G | 21.68174 | 0.20465 | Preparative yield |
|  |  | GML+PCA | Suggestive | 5 | 1,883,165 | 4.6E−05 | A/G | 18.57534 | 0.14972 | Preparative yield |
| 6 | PotVar0034580 | GLM | FDR (0.05) | 5 | 51,697,927 | 2.1E−05 | T/C | 19.33918 | 0.18535 | Preparative yield |
|  |  | GLM+Q | FDR (0.05) | 5 | 51,697,927 | 2.09E−05 | T/C | 20.39375 | 0.19744 | Preparative yield |
| 7 | solcap_snp_c1_1250 | GLM | FDR (0.05) | 5 | 51,978,384 | 4.94E−06 | A/C | 23.69015 | 0.21211 | Preparative yield |
|  |  | GLM+Q | FDR (0.05) | 5 | 51,978,384 | 4.09E−06 | A/C | 24.26386 | 0.22014 | Preparative yield |
| 8 | solcap_snp_c2_3174 | GLM | FDR (0.05) | 6 | 4,419,112 | 8.6E−06 | A/G | 22.41938 | 0.20678 | Preparative yield |
|  |  | GLM+Q | FDR (0.05) | 6 | 4,419,112 | 1.22E−05 | A/G | 21.65385 | 0.20504 | Preparative yield |
| 9 | solcap_snp_c1_5970 | GLM+Q | FDR (0.05) | 7 | 54,862,076 | 4.9E−05 | T/G | 18.32787 | 0.17771 | Preparative yield |
| 10 | PotVar0012073 | GLM | FDR (0.05) | 9 | 2,679,615 | 1.2E−05 | A/G | 19.87446 | 0.18596 | Preparative yield |
|  |  | GLM+Q | FDR (0.05) | 9 | 2,679,615 | 1.17E−05 | A/G | 21.73143 | 0.20345 | Preparative yield |
| 11 | solcap_snp_c2_5957 | GLM+Q | FDR (0.05) | 11 | 2,855,044 | 5E−05 | A/G | 18.35478 | 0.18175 | Preparative yield |
| 12 | solcap_snp_c2_6309 | GLM | FDR (0.05) | 11 | 2,979,784 | 1.2E−05 | T/C | 21.81045 | 0.21214 | Preparative yield |
|  |  | GLM+Q | FDR (0.05) | 11 | 2,979,784 | 1.64E−05 | T/C | 21.11726 | 0.20921 | Preparative yield |
| 13 | PotVar0067347 | GLM | FDR (0.05) | 11 | 2,973,509 | 2.1E−05 | T/C | 20.28743 | 0.19087 | Preparative yield |
|  |  | GLM+Q | FDR (0.05) | 11 | 2,973,509 | 2.29E−05 | T/C | 20.14393 | 0.19353 | Preparative yield |
| 14 | solcap_snp_c2_6285 | GLM | FDR (0.05) | 11 | 3,080,240 | 8.2E−06 | A/G | 22.48715 | 0.20539 | Preparative yield |
|  |  | GLM+Q | FDR (0.05) | 11 | 3,080,240 | 5.57E−06 | A/G | 23.54087 | 0.21734 | Preparative yield |
| 15 | solcap_snp_c2_6185 | GLM | FDR (0.05) | 11 | 3,572,445 | 2.5E−05 | A/C | 19.87485 | 0.18772 | Preparative yield |
|  |  | GLM+Q | FDR (0.05) | 11 | 3,572,445 | 3.37E−05 | A/C | 19.23402 | 0.1859 | Preparative yield |
| 16 | PotVar0120075 | GLM+Q | FDR (0.05) | 1 | 433,696 | 5.9E−05 | A/G | 17.91699 | 0.16272 | Aspect ratio |
|  |  | GLM+Q | FDR (0.05) | 1 | 433,696 | 8.1E−05 | A/G | 17.16788 | 0.15815 | Roundness |
| 17 | PotVar0119973 | GLM+Q | FDR (0.05) | 1 | 472,422 | 6.7E−05 | T/C | 17.60949 | 0.16041 | Aspect ratio |
|  |  | GLM+Q | FDR (0.05) | 1 | 472,422 | 2.1E−05 | T/C | 20.3393 | 0.18167 | Roundness |
| 18 | PotVar0119913 | GLM+Q | FDR (0.05) | 1 | 473,487 | 6.7E−05 | A/G | 17.60949 | 0.16041 | Aspect ratio |
|  |  | GLM+Q | FDR (0.05) | 1 | 473,487 | 2.1E−05 | A/G | 20.3393 | 0.18167 | Roundness |
| 19 | solcap_snp_c2_36659 | GLM+Q | FDR (0.05) | 1 | 532,003 | 5.4E−05 | T/C | 18.11193 | 0.16418 | Aspect ratio |
|  |  | GLM+Q | FDR (0.05) | 1 | 532,003 | 7.3E−05 | T/C | 17.4212 | 0.16008 | Roundness |
| 20 | solcap_snp_c2_36664 | GLM+Q | FDR (0.05) | 1 | 535,454 | 8.9E−05 | A/G | 16.95617 | 0.15717 | Aspect ratio |
|  |  | GLM+Q | FDR (0.05) | 1 | 535,454 | 9.6E−05 | A/G | 16.77846 | 0.1568 | Roundness |
| 21 | solcap_snp_c2_36665 | GLM+Q | FDR (0.05) | 1 | 536,033 | 3.4E−05 | A/G | 19.20315 | 0.17223 | Aspect ratio |
|  |  | GLM+Q | FDR (0.05) | 1 | 536,033 | 3.7E−05 | A/G | 18.96398 | 0.17165 | Roundness |

| N | SNP | Statistical model | Significance levels | Chr | Position | p-Value* | Polymorphism | QTL effect | Heritability value | Trait |
|---|---|---|---|---|---|---|---|---|---|---|
| 22 | PotVar0071846 | GLM+Q | FDR (0.05) | 1 | 1,155,603 | 3.3E−05 | A/C | 19.20053 | 0.17141 | Aspect ratio |
| | | GLM+Q | FDR (0.05) | 1 | 1,155,603 | 2.5E−05 | A/C | 19.87454 | 0.17799 | Roundness |
| 23 | PotVar0071852 | GLM+Q | FDR (0.05) | 1 | 1,155,770 | 3.3E−05 | T/C | 19.26069 | 0.17265 | Aspect ratio |
| | | GLM+PCA | Suggestive | 1 | 1,155,770 | 8.8E−05 | T/C | 17.02399 | 0.15571 | Aspect ratio |
| | | GLM+Q | FDR (0.05) | 1 | 1,155,770 | 3.5E−05 | T/C | 19.1343 | 0.17291 | Roundness |
| 24 | solcap_snp_c2_56617 | GLM | FDR (0.05) | 2 | 4,833,289 | 5.5E−05 | A/G | 17.96239 | 0.16952 | Aspect ratio |
| | | GLM | FDR (0.05) | 2 | 4,833,289 | 6.3E−05 | A/G | 17.67625 | 0.16727 | Roundness |
| 25 | solcap_snp_c1_16379 | GLM | Bonferroni | 2 | 4,839,524 | 9.2E−07 | T/C | 28.02326 | 0.24794 | Aspect ratio |
| | | GLM+Q | FDR (0.05) | 2 | 4,839,524 | 5.4E−05 | T/C | 18.17052 | 0.16451 | Aspect ratio |
| | | GLM+PCA | Suggestive | 2 | 4,839,524 | 6.4E−05 | T/C | 17.81461 | 0.15835 | Aspect ratio |
| | | GLM | Bonferroni | 2 | 4,839,524 | 9.2E−07 | T/C | 27.29335 | 0.24305 | Roundness |
| | | GLM+Q | FDR (0.05) | 2 | 4,839,524 | 2.8E−05 | T/C | 19.73863 | 0.17492 | Roundness |
| | | GLM+PCA | Suggestive | 2 | 4,839,524 | 3E−05 | T/C | 19.58447 | 0.17361 | Roundness |
| 26 | PotVar0032432 | GLM | FDR (0.05) | 2 | 5,126,968 | 2.7E−05 | A/G | 19.76242 | 0.19231 | Aspect ratio |
| | | GLM | FDR (0.05) | 2 | 5,126,968 | 2.7E−05 | A/G | 19.28304 | 0.18853 | Roundness |
| 27 | PotVar0032402 | GLM | FDR (0.05) | 2 | 5,128,062 | 9.6E−06 | A/G | 22.14307 | 0.20476 | Aspect ratio |
| | | GLM | FDR (0.05) | 2 | 5,128,062 | 9.9E−06 | A/G | 22.07346 | 0.20424 | Roundness |
| 28 | solcap_snp_c2_4353 | GLM | FDR (0.05) | 2 | 5,158,305 | 7.9E−06 | T/C | 22.64938 | 0.2104 | Aspect ratio |
| | | GLM | FDR (0.05) | 2 | 5,158,305 | 8.7E−06 | T/C | 22.42338 | 0.20874 | Roundness |
| 29 | solcap_snp_c2_4354 | GLM | FDR (0.05) | 2 | 5,159,301 | 2.2E−05 | T/C | 20.07171 | 0.18573 | Aspect ratio |
| | | GLM | FDR (0.05) | 2 | 5,159,301 | 2.5E−05 | T/C | 19.85727 | 0.18411 | Roundness |
| 30 | PotVar0032114 | GLM | FDR (0.05) | 2 | 5,159,377 | 1.9E−05 | A/C | 20.51208 | 0.19258 | Aspect ratio |
| | | GLM | FDR (0.05) | 2 | 5,159,377 | 1.9E−05 | A/C | 20.51208 | 0.19258 | Roundness |
| 31 | solcap_snp_c2_4360 | GLM | Bonferroni | 2 | 5,356,498 | 6.7E−07 | A/G | 28.68696 | 0.24585 | Aspect ratio |
| | | GLM+Q | FDR (0.05) | 2 | 5,356,498 | 2.1E−05 | A/G | 20.2877 | 0.17924 | Aspect ratio |
| | | GLM+PCA | Suggestive | 2 | 5,356,498 | 1.5E−05 | A/G | 21.20211 | 0.18928 | Aspect ratio |
| | | GLM | Bonferroni | 2 | 5,356,498 | 6.7E−07 | A/G | 27.44411 | 0.23773 | Roundness |
| | | GLM+Q | FDR (0.05) | 2 | 5,356,498 | 9.6E−06 | A/G | 22.18673 | 0.19441 | Roundness |
| | | GLM+PCA | Suggestive | 2 | 5,356,498 | 1.5E−05 | A/G | 21.20211 | 0.18928 | Roundness |
| 32 | solcap_snp_c2_57190 | GLM | FDR (0.05) | 2 | 5,394,420 | 1.7E−05 | A/C | 20.70086 | 0.19221 | Aspect ratio |
| | | GLM | FDR (0.05) | 2 | 5,394,420 | 1.8E−05 | A/C | 20.55906 | 0.19114 | Roundness |
| 33 | solcap_snp_c2_48725 | GLM | FDR (0.05) | 2 | 5,510,774 | 2.2E−05 | T/C | 20.07171 | 0.18573 | Aspect ratio |
| | | GLM | FDR (0.05) | 2 | 5,510,774 | 2.5E−05 | T/C | 19.85727 | 0.18411 | Roundness |
| 34 | solcap_snp_c1_1503 | GLM | FDR (0.05) | 2 | 5,362,501 | 2.4E−05 | T/C | 19.91747 | 0.18629 | Aspect ratio |
| | | GLM | FDR (0.05) | 2 | 5,362,501 | 2.4E−05 | T/C | 19.91934 | 0.1863 | Roundness |
| 35 | solcap_snp_c2_48735 | GLM | FDR (0.05) | 2 | 5,552,679 | 2.4E−05 | A/G | 19.91747 | 0.18629 | Aspect ratio |
| | | GLM | FDR (0.05) | 2 | 5,55,2679 | 2.4E−05 | A/G | 19.91934 | 0.1863 | Roundness |
| 36 | solcap_snp_c1_3750 | GLM | FDR (0.05) | 2 | 6,090,074 | 4.2E−06 | T/C | 24.121 | 0.21707 | Aspect ratio |
| | | GLM | FDR (0.05) | 2 | 6,090,074 | 9E−06 | T/C | 22.27071 | 0.20381 | Roundness |

| N | SNP | Statistical model | Significance levels | Chr | Position | p-Value* | Polymorphism | QTL effect | Heritability value | Trait |
|---|---|---|---|---|---|---|---|---|---|---|
| 37 | solcap_snp_c1_16405 | GLM | FDR (0.05) | 2 | 6,707,427 | 6.6E−06 | A/G | 23.03216 | 0.20932 | Aspect ratio |
| | | GLM+PCA | Suggestive | 2 | 6,707,427 | 9.6E−05 | A/G | 16.83224 | 0.15409 | Aspect ratio |
| | | GLM | FDR (0.05) | 2 | 6,707,427 | 6.6E−06 | A/G | 23.89646 | 0.21548 | Roundness |
| | | GLM+PCA | Suggestive | 2 | 6,707,427 | 5E−05 | A/G | 18.34947 | 0.17054 | Roundness |
| 38 | solcap_snp_c1_3746 | GLM | FDR (0.05) | 2 | 7,050,595 | 1.5E−05 | T/C | 21.04094 | 0.19657 | Aspect ratio |
| | | GLM | FDR (0.05) | 2 | 7,050,595 | 1.4E−05 | T/C | 21.24232 | 0.19808 | Roundness |
| 39 | solcap_snp_c1_3747 | GLM | Bonferroni | 2 | 7,050,714 | 1.9E−06 | T/C | 26.12617 | 0.23301 | Aspect ratio |
| | | GLM+Q | FDR (0.05) | 2 | 7,050,714 | 3.2E−05 | T/C | 19.37954 | 0.17319 | Aspect ratio |
| | | GLM+PCA | Suggestive | 2 | 7,050,714 | 2.2E−05 | T/C | 20.28852 | 0.17686 | Aspect ratio |
| | | GLM | Bonferroni | 2 | 7,050,714 | 1.9E−06 | T/C | 27.64594 | 0.24326 | Roundness |
| | | GLM+Q | FDR (0.05) | 2 | 7,050,714 | 2.2E−05 | T/C | 20.28477 | 0.17974 | Roundness |
| | | GLM+PCA | Suggestive | 2 | 7,050,714 | 8.6E−06 | T/C | 22.58985 | 0.19625 | Roundness |
| 40 | solcap_snp_c2_14652 | GLM | Bonferroni | 2 | 8,703,476 | 1.7E−06 | A/G | 26.35701 | 0.23458 | Aspect ratio |
| | | GLM+Q | FDR (0.05) | 2 | 8,703,476 | 5.1E−05 | A/G | 18.28407 | 0.16778 | Aspect ratio |
| | | GLM+PCA | Suggestive | 2 | 8,703,476 | 6.5E−05 | A/G | 17.74889 | 0.16164 | Aspect ratio |
| | | GLM | Bonferroni | 2 | 8,703,476 | 1.7E−06 | A/G | 24.01768 | 0.21831 | Roundness |
| | | GLM+Q | FDR (0.05) | 2 | 8,703,476 | 4.4E−05 | A/G | 18.62552 | 0.17228 | Roundness |
| | | GLM+PCA | Suggestive | 2 | 8,703,476 | 4.4E−05 | A/G | 18.65739 | 0.17427 | Roundness |
| 41 | solcap_snp_c2_14648 | GLM | Bonferroni | 2 | 8,799,527 | 1.4E−06 | A/C | 27.01056 | 0.24553 | Aspect ratio |
| | | GLM+Q | FDR (0.05) | 2 | 8,799,527 | 4.9E−05 | A/C | 18.45696 | 0.17279 | Aspect ratio |
| | | GLM+PCA | Suggestive | 2 | 8,799,527 | 1E−04 | A/C | 16.83119 | 0.15898 | Aspect ratio |
| | | GLM | Bonferroni | 2 | 8,799,527 | 1.4E−06 | A/C | 24.68137 | 0.22921 | Roundness |
| | | GLM+Q | FDR (0.05) | 2 | 8,799,527 | 4.8E−05 | A/C | 18.48983 | 0.17371 | Roundness |
| | | GLM+PCA | Suggestive | 2 | 8,799,527 | 6.7E−05 | A/C | 17.74091 | 0.17059 | Roundness |
| 42 | solcap_snp_c2_32254 | GLM | FDR (0.05) | 2 | 13,697,523 | 4.9E−05 | A/G | 18.29319 | 0.1754 | Aspect ratio |
| | | GLM | FDR (0.05) | 2 | 13,697,523 | 4.9E−05 | A/G | 18.82751 | 0.1796 | Roundness |
| 43 | PotVar0022442 | GLM | FDR (0.05) | 7 | 1,120,011 | 7.4E−05 | T/C | 17.33025 | 0.16772 | Aspect ratio |
| | | GLM | FDR (0.05) | 7 | 1,120,011 | 1.94E−04 | T/C | 15.16808 | 0.14993 | Roundness |
| 44 | solcap_snp_c2_33657 | GLM | FDR (0.05) | 11 | 2,274,063 | 4.4E−05 | A/G | 18.55125 | 0.17744 | Aspect ratio |
| | | GLM | FDR (0.05) | 11 | 2,274,063 | 4.00E−05 | A/G | 18.75715 | 0.17905 | Roundness |
| 45 | solcap_snp_c1_1504 | GLM | FDR (0.05) | 2 | 4,479,389 | 6.5E−06 | A/G | 23.10075 | 0.21174 | Aspect ratio |
| | | GLM | FDR (0.05) | 2 | 4,479,389 | 1.36E−05 | A/G | 21.29554 | 0.19848 | Roundness |
| 46 | solcap_snp_c2_50824 | GLM+Q | FDR (0.05) | 12 | 54,479,773 | 2.7E−05 | T/C | 20.01852 | 0.18645 | Aspect ratio |
| | | GLM+Q | FDR (0.05) | 12 | 54,479,773 | 6.56E−05 | T/C | 17.88054 | 0.17686 | Circularity |
| | | GLM+Q | FDR (0.05) | 12 | 54,479,773 | 2.78E−04 | T/C | 14.54952 | 0.14453 | Roundness |
| 47 | solcap_snp_c2_54815 | GLM+Q | FDR (0.05) | 1 | 1,125,639 | 9.1E−05 | A/G | 16.98376 | 0.16372 | Aspect ratio |
| | | GLM+Q | FDR (0.05) | 1 | 1,125,639 | 6.3E−05 | A/G | 17.82237 | 0.17128 | Roundness |
| 48 | solcap_snp_c1_15462 | GLM+Q | Bonferroni | 7 | 205,949 | 1.3E−06 | A/G | 27.20638 | 0.23095 | 1st bi-component |
| | | GLM+PCA | Suggestive | 7 | 205,949 | 1.3E−05 | A/G | 21.51624 | 0.17525 | 1st bi-component |
| | | GLM+Q | FDR (0.05) | 7 | 205,949 | 5.6E−06 | A/G | 23.5453 | 0.20251 | Circularity |
| | | GLM+PCA | Suggestive | 7 | 205,949 | 5E−05 | A/G | 18.37838 | 0.1726 | Circularity |

 

| N | SNP | Statistical model | Significance levels | Chr | Position | p-Value* | Polymorphism | QTL effect | Heritability value | Trait |
|---|---|---|---|---|---|---|---|---|---|---|
| 49 | PotVar0047235 | GLM+PCA | FDR (0.05) | 11 | 39,417,958 | 5.7E−06 | T/C | 23.51133 | 0.20444 | Circularity |
| 50 | solcap_snp_c2_52067 | GLM | Suggestive Suggestive | 5 | 1,847,556 | 3.7E−05 | A/G | 18.89164 | 0.17674 | 1st bi-component |
| | | GLM+Q | FDR (0.05) | 5 | 1,847,556 | 8.8E−05 | A/G | 16.97574 | 0.15589 | 1st bi-component |
| 51 | PotVar0097020 | GLM+Q | FDR (0.05) | 9 | 60,563,436 | 4.3E−06 | T/G | 24.1761 | 0.21034 | 1st bi-component |
| | | GLM+PCA | Suggestive | 9 | 60,563,436 | 1.8E−05 | T/G | 20.69739 | 0.1683 | 1st bi-component |
| 52 | PotVar0097065 | GLM+Q | Bonferroni | 9 | 60,565,011 | 2.8E−06 | T/C | 25.31005 | 0.22143 | 1st bi-component |
| | | GLM+PCA | Suggestive | 9 | 60,565,011 | 2.6E−05 | T/G | 19.92614 | 0.16731 | 1st bi-component |
| 53 | PotVar0052560 | GLM | Suggestive | 12 | 59,793,471 | 6.3E−05 | A/G | 17.68548A | 0.16894 | 1st bi-component |

**Note:**
* Only minimal $p$-value for each SNP presented.

*Aspect ratio*

The most promising results obtained for morphological traits related not to size but to shape of the starch granules. GLM without population structure analysis yielded five SNPs with (−lg $p$)-values exceeding the Bonferroni level (Table 1; Fig. 2). All of these SNPs are tightly grouped on chromosome 2, forming qualitative trait loci (QTL). Four other noticeable SNPs determined by GLM have (−lg $p$)-values exceeding FDR level and are located on chromosomes 1, 7, 11 and 12. The same three SNPs were confirmed by the GLM+PCA method and are probably parts of appropriate QTLs. Belonging to QTLs increases the probability that certain SNPs are associated with the trait studied.

*Roundness*

Analysis of SNPs for the "Roundness" trait in detail reproduced the results obtained for the "Aspect ratio". Table 1 contains SNPs associated with the "Aspect ratio" and includes SNPs for "Roundness" as well. Manhattan plots are highly similar for the two traits because both describe the shape of potato starch granules.

*Circularity*

Three SNPs associated with the "Circularity" trait were revealed by GWAS using the statistical models GLM+PCA and GLM+Q (Table 1; Fig. S9).

*SNPs correlated with the first phenotypic bicomponent*

Since the "Preparative yield", "Area", "Circularity", "Feret's diameter", "Minimal Feret's diameter" and "Solidity" traits mutually contribute the first principal component significantly (Table S3), it was logical to analyze SNPs' associations with the first bicomponent to determine if there were any SNPs associated with all of the traits simultaneously.

One of the SNPs found for the first bicomponent (solcap_snp_c1_15462 on the seventh chromosome) fits the SNP, which is significant for the "Circularity" trait. The "Circularity" trait contributes the first bicomponent (−0.67992). The other four significant SNPs are unique and not associated with any single trait. All of these SNPs are a part of appropriate

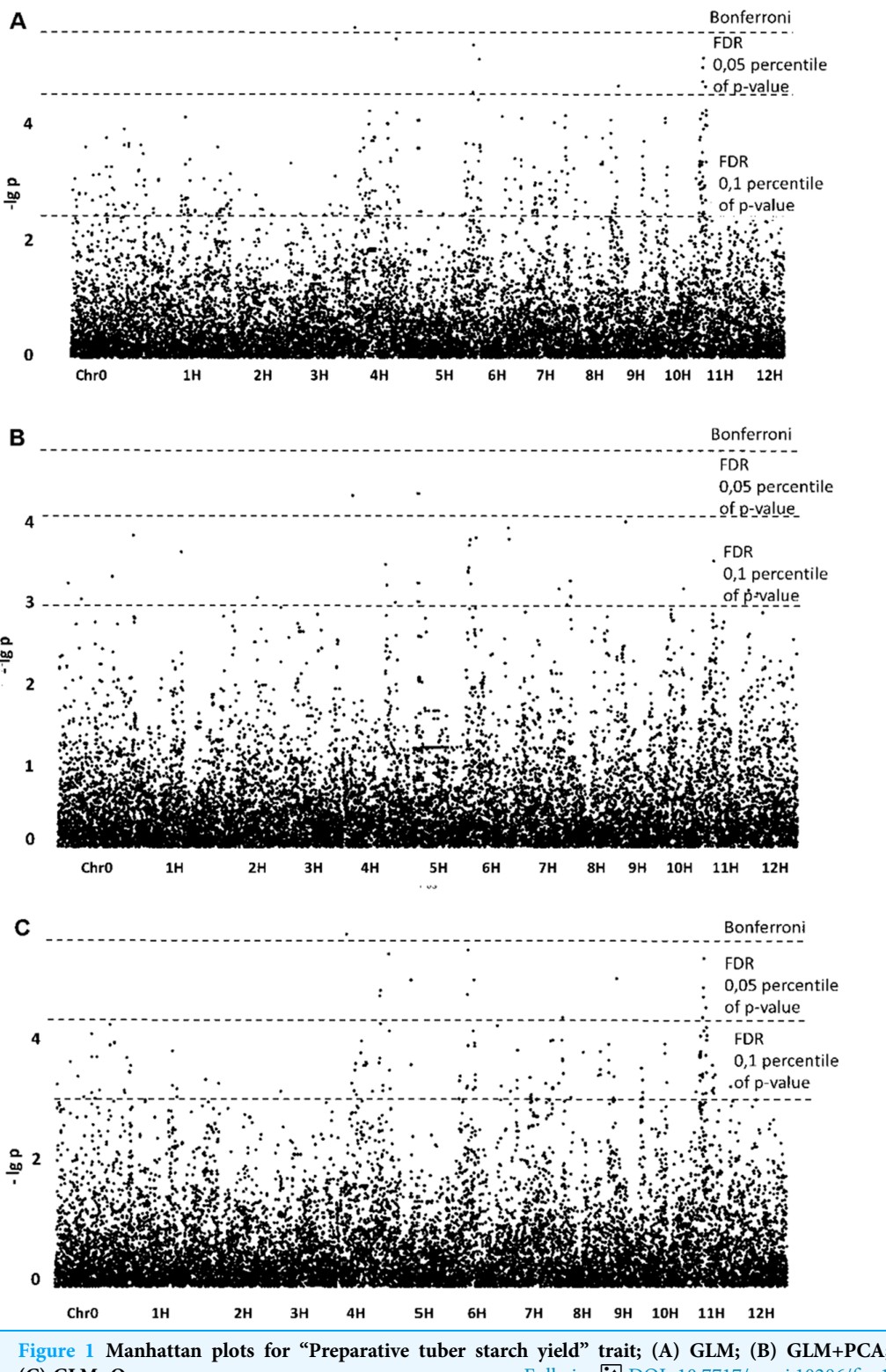

**Figure 1 Manhattan plots for "Preparative tuber starch yield" trait; (A) GLM; (B) GLM+PCA; (C) GLM+Q.**

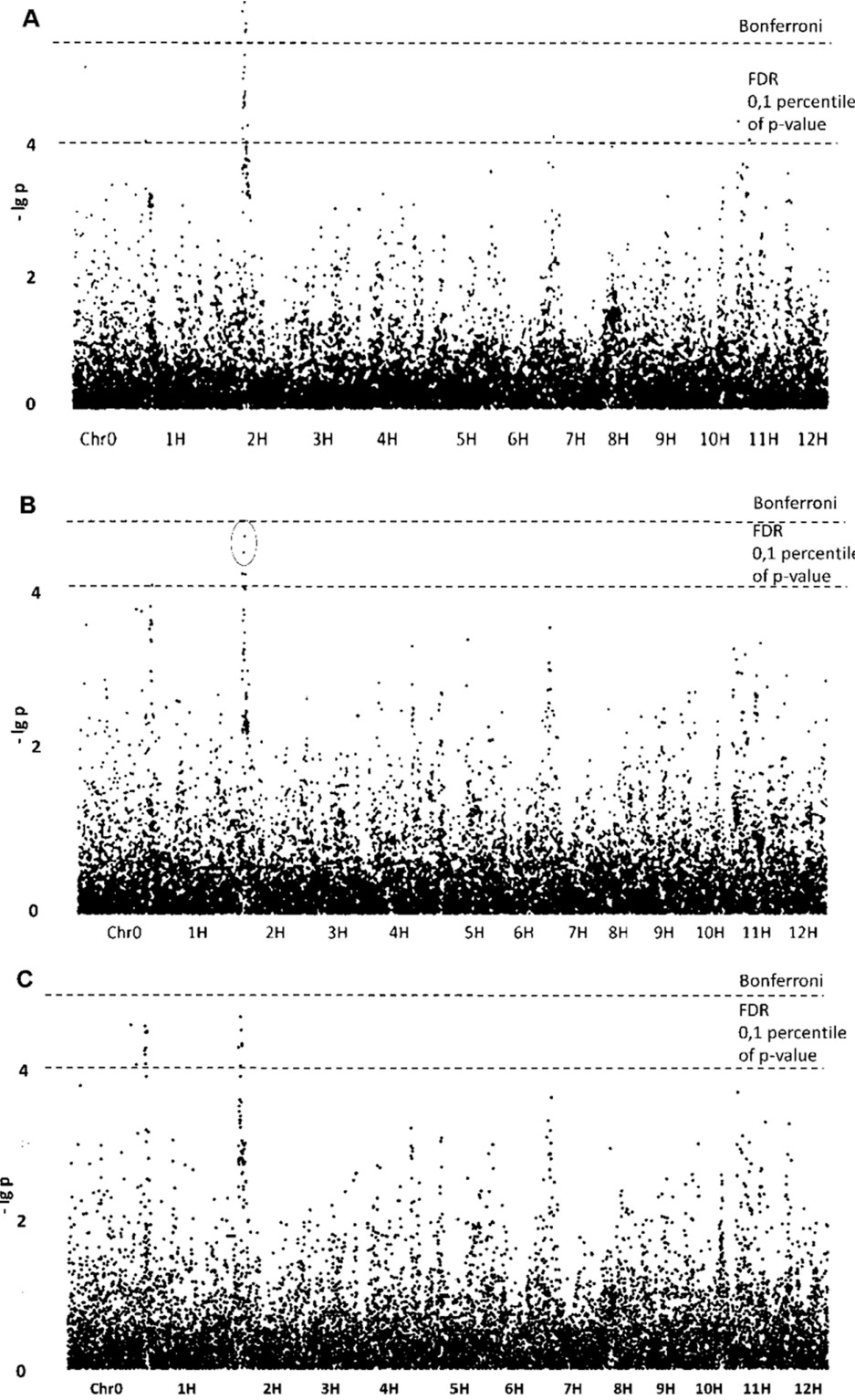

**Figure 2 Manhattan plots for "Aspect Ratio" trait. (A) GLM; (B) GLM+PCA; (C) GLM+Q.**

QTLs ("supported" with other SNPs with lower (−lg $p$)-values) (Fig. S10). Genes around the SNPs may be responsible for the general starch carbohydrate polymer arrangement in the starch granule.

No significant SNPs associated with the second and third bicomponents were revealed.

## SNPS AND RELATED GENES/PROTEINS

In total, 53 SNPs were found to be significant in the association study of the potato genotypes and starch morphology/yield-related traits. (−lg $p$)-Values of one SNP were associated with "preparative yield of starch", five SNPs—with "aspect ratio" and "roundness" traits and one SNP—with the first bicomponent exceeded strict Bonferroni criteria. Nevertheless, some associated SNPs overcame 0.1 and 0.05 FDR (false discovery rate) and suggestive levels. Most of the SNPs are located on the first and second chromosomes, the least on chromosomes 4, 5, 6, 7, 9, 11 and 12. A total of 37 of 53 SNPs are located in protein coding regions. Not one of the SNPs studied in this paper coincides with the ones associated with covalently bound phosphorus content in starch (*Khlestkin et al., 2019*) (Table S4).

## PREPARATIVE YIELD OF STARCH

Three out of four SNPs found on chromosome 4 are located in noncoding sequences and relate to two different DNA regions. There are some other related SNPs with considerably lower significance in these regions, indicating possible QTLs. The fourth SNP is situated in the different region of the gene, coding for a DNA transcription regulator consisting of 95 amino acid residues. All three SNPs found on chromosome 5 are related to the sequences coding different proteins. One SNP corresponds to coiled coil protein with unclear function; two others are related to low-molecular-weight organic metabolite conversion, specifically to the enzymes aldehyde dehydrogenase and phenyl alanine lyase, which are responsible for alcohol-aldehyde equilibria in the cell and for flavonoid synthesis. Taking into account SNPs that are less significant but still associated with starch yield, we may speak about two trait-associated DNA regions that are potential QTLs. The significant SNP on chromosome 6 seems also be a part of a QTL and relates to plastid transcriptionally active protein 16 (PTAC16), which is suspected to be involved in the regulation of plastid gene expression. The significant SNP on chromosome 7 encodes the protein translation factor SUI1. The significant SNP on chromosome 9 encodes a protein with unknown function. A group of five SNPs located within 2855044–3572445 bp on chromosome 11 belongs to the same QTL, and the SNPs are included in the coding regions of several proteins: DEGP10, methylenetetrahydrofolate reductase, acetolactate synthase, and two proteins of unknown function. In general, most of the proteins associated with potato starch preparative yield variations are involved in plastid activity and low-molecular-weight metabolite biosynthesis.

## ASPECT RATIO AND ROUNDNESS

These traits are well-correlated with each other and describe the shape of starch granules in a similar way; thus, the traits are associated with the same SNPs. There is a locus on

chromosome 1 containing eight SNPs in the same DNA region. Some of these proteins are related to circadian rhythm-regulating proteins. The circadian clock regulates numerous plant developmental and metabolic processes. In crop species, the circadian clock contributes significantly to plant performance and productivity and to the adaptation and geographical range over which crops can be grown. Other SNPs are related to the phosphorylation of important biochemical intermediates and plastid organization. On chromosome 2, a total of 19 significant SNPs were identified. The SNPs are narrowly situated, forming a single DNA region, a potential trait-related QTL. Most of the SNPs are located in noncoding regions or related to proteins with unknown functions. Two SNPs are related to WPP domain-associated protein-encoding genomic regions, one to plastid high chlorophyll fluorescence 136, another to pentatricopeptide repeat-containing protein, and one SNP is related to DNA binding protein. Two SNPs on chromosomes 7 and 11 were found in the genes encoding proteins with unknown functions.

## CIRCULARITY

Only three SNPs associated with the circularity trait were identified. The first SNP is related to a noncoding region of chromosome 12, the second is located on chromosome 7 in the minichromosome maintenance 5 protein coding sequence, and the third is located on chromosome 11 in the flavonoid 3′,5′-hydroxylase-encoding region.

## FIRST "PHENOTYPE" BICOMPONENT

Six SNPs were found to be associated with the first bicomponent from 2B-PLS, which is a complex component comprising all of the traits studied. Thus, one SNP was the same for the first bicomponent and Preparative yield (chromosome 7), while another SNP was the same for the first bicomponent and Circularity (chromosome 5). Among the four other unique 1st bicomponent associated SNPs, two are related to the GWD gene, one of the key starch biosynthesis genes (*Khlestkin, Peltek & Kolchanov, 2017*), which is responsible for the phosphorylation-dephosphorylation of glucans (chromosome 9). The other two SNPs are associated with cytochrome B561 family protein (chromosome 5) and Pto-interacting protein 1 (chromosome 12).

In summary, chromosomes 1 and 2 contain important regions responsible for the roundness and aspect ratio of tuber starch granules. There are indications that granule shape may depend on circadian rhythm-related metabolic processes and starch phosphorylation processes. The GWD gene, which is known to regulate phosphorylation and dephosphorylation participates in the regulation of a whole number of morphological traits, rather than a single certain one. Nevertheless, some other mechanisms and proteins located on chromosome 2 influence the granule formation process. The preparative yield of tuber starch is probably a polygenic trait, regulated by a number of proteins that are encoded by sequences in various parts and chromosomes of the potato genome.

## CONCLUSIONS

A genome-wide association study using a 22K SNP potato array enabled 53 novel SNPs to be identified on chromosomes 1, 2, 4, 5, 6, 7, 9, 11 and 12; these SNPs are associated with tuber starch preparative yield and with starch granule morphology (aspect ratio, roundness, circularity, and the first bicomponent). Some of the SNPs observed in this study are located in noncoding regions. The coding regions are associated with membrane and plastid proteins, DNA transcription and binding regulators, low-molecular-weight metabolite synthesis as well as flavonoid biosynthesis. The information on significant regions can be used to convert SNPs to PCR-markers, convenient for screening breeding material in programs aimed on development of potato varieties with desired starch properties.

## ACKNOWLEDGEMENTS

We thank ICG collection "GenAgro" (Novosibirsk, Russia) providing potato plant collection and personally Anna Safonova. We also thank the Traitgenetics GmbH (Gatersleben, Germany) for providing service on genotyping of potato cultivars using 22K Illumina SNP array.

### Funding

This work was supported by the Russian Foundation for Basic Research (No 17-29-08006). GenAgro plant collection is supported by ICG (Project № 0324-2019-0039-C-01, AAAA-A16-116061750188-4). The funders had no role in study design, data collection and analysis, decision to publish, or preparation of the manuscript.

### Grant Disclosures

The following grant information was disclosed by the authors:
Russian Foundation for Basic Research: 17-29-08006.
ICG: 0324-2019-0039-C-01 and AAAA-A16-116061750188-4.

### Competing Interests

The authors declare that they have no competing interests. Patent application sited (de Vetten NCMH, Heeres P. Method for modifying the size and/or morphology of starch granules. 2004. EP 1473307A1. European Patent Office) has "withdrown" status.

### Author Contributions

- Vadim K. Khlestkin conceived and designed the experiments, performed the experiments, analyzed the data, prepared figures and/or tables, authored or reviewed drafts of the paper, he is an author of the main idea of the paper to regulate practically important starch properties at gene level, and approved the final draft.
- Tatyana V. Erst performed the experiments, authored or reviewed drafts of the paper, and approved the final draft.

- Irina V. Rozanova analyzed the data, prepared figures and/or tables, and approved the final draft.
- Vadim M. Efimov analyzed the data, prepared figures and/or tables, and approved the final draft.
- Elena K. Khlestkina conceived and designed the experiments, authored or reviewed drafts of the paper, teacher, and approved the final draft.

## Patent Disclosures

The following patent dependencies were disclosed by the authors:

De Vetten N.C.M.H, Heeres P. Method for modifying the size and/or morphology of starch granules. 2004. EP 1473307A1. European Patent Office.

Authors of this article do not own the patent. The patent application currently has a "withdrawn" status.

## Data Availability

Genotyping data and raw data on the mean values are available in the Supplemental Files.

## Supplemental Information

Supplemental information for this article can be found online at http://dx.doi.org/10.7717/peerj.10286#supplemental-information.

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
