# Peer review of "Genetic loci determining potato starch yield and granule morphology revealed by genome-wide association study (GWAS)"

_PeerJ, doi:10.7717/peerj.10286_

## Round 0.1 · original submission · Minor Revisions

Authors need to consider the comments of reviewers when preparing a new version of the manuscript.

Reviewer 1 ·

Basic reporting

See General comments

Experimental design

See General comments

Validity of the findings

See General comments

Additional comments

The paper is on the genetic loci determination related to starch granules in potato. A total of 90 potato samples were used. GWD gene is related to the results. I believe that the paper is interesting and showed new insights in to starch biosynthesis in potatoes. i only have few comments and rather suggest the paper to be acceptance after making comments to mine.
There are ;already papers on the genes responsible for the granule morphology in different crops such as arabidopsis. I hope the authors may comment on how the findings from other crops can be related to the potatoes.
Figure 1, the captions should be translated as most of the readers don't learn Russian. also, which year is this drawn?
GWAS in the title, this may be spelled out in full.
How about the number of starch granules in the amyloplasts?
There are still some errors in your writings, Please double check your paper again to make sure not many mistakes are there.

Reviewer 2 ·

Basic reporting

1. As I am from a non-English speaking country, it is difficult to judge the English of the manuscript.
2. I feel that in the section Results and Discussion, the literature referencing is insufficient. For instance, the authors discuss the connection of identified SNPs to certain proteins associated with studied traits. However, no references were provided neither for proteins itself nor for the hypothesis behind putative associations.
3. In the section Materials, it is not clear why the authors did not provide a full list of accessions in the study? A simple reference to another published work is not sufficient. Please provide a separate file with the list of accessions, their origin, and other details.
4. It is not clear why the MLM method was not applied in the GWAS, although the method was stated in the section “Materials and Methods”? A small-sized population and the GLM (alone or with a combination of matrices) are often lead to the detection of false-positive signals due to inflated p-value.
5. I think that Table S4 should be moved to the text itself. The Table itself should have more information, such as QTL effect and trait heritability values. In figures with Manhattan plots, I think that all figs should be given with side-by-side QQ plots.

Experimental design

I have no question to this section, except the fact that the studied population is rather small-sized, which may lead to potential false statements in QTN identification for studied traits.

Validity of the findings

The results of the study are fairly novel. However, I recommend the authors use not only GLM but also the MLM method, which is a more reliable approach to avoid the detection of possible false-positive associations.
Conclusions should be reassessed after the application of MLM methods in the GWAS.

Additional comments

Table S4 is missing information for the QTL effects of the associations.
Also, please provide heritability values for analyzed traits in the study?

Reviewer 3 ·

Basic reporting

Ok

Experimental design

OK

Validity of the findings

OK

Additional comments

1. Fig.1 should have english note beside the Russian;
2. Only one environment for the trait phenotyping, can this make the significant SNP reliable;
3. Move Some figures and tables to main text from the supplimentary file;
4. Is there a candidate gene on Chr.2H for the siginficant SNP or a published QTL for this region;

---

## Round 0.2 · Minor Revisions

Authors need to provide the following information on the material under study:

- The genotype information for each line needs to be made available as a supplemental data set.

- There is little information about the experimental design.

- How were the plants grown? field or greenhouse? what was the layout? how old when harvested? what time of year? what was the replication?

---

## Round 0.3 · accepted · Accept

The manuscript is ready for publication, all comments from the reviewers have been addressed.